# Reasoning With Neural Tensor Networks for Knowledge Base Completion

**Richard Socher,**\* **Danqi Chen\*, Christopher D. Manning, Andrew Y. Ng**
Computer Science Department, Stanford University, Stanford, CA 94305, USA
richard@socher.org, {danqi,manning}@stanford.edu, ang@cs.stanford.edu

## Abstract

Knowledge bases are an important resource for question answering and other tasks but often suffer from incompleteness and lack of ability to reason over their discrete entities and relationships. In this paper we introduce an expressive neural tensor network suitable for reasoning over relationships between two entities. Previous work represented entities as either discrete atomic units or with a single entity vector representation. We show that performance can be improved when entities are represented as an average of their constituting word vectors. This allows sharing of statistical strength between, for instance, facts involving the "Sumatran tiger" and "Bengal tiger." Lastly, we demonstrate that all models improve when these word vectors are initialized with vectors learned from unsupervised large corpora. We assess the model by considering the problem of predicting additional true relations between entities given a subset of the knowledge base. Our model outperforms previous models and can classify unseen relationships in WordNet and FreeBase with an accuracy of 86.2% and 90.0%, respectively.

## 1  Introduction

Ontologies and knowledge bases such as WordNet [1], Yago [2] or the Google Knowledge Graph are extremely useful resources for query expansion [3], coreference resolution [4], question answering (Siri), information retrieval or providing structured knowledge to users. However, they suffer from incompleteness and a lack of reasoning capability.

Much work has focused on extending existing knowledge bases using patterns or classifiers applied to large text corpora. However, not all common knowledge that is obvious to people is expressed in text [5, 6, 2, 7]. We adopt here the complementary goal of predicting the likely truth of additional facts based on existing facts in the knowledge base. Such factual, common sense reasoning is available and useful to people. For instance, when told that a new species of monkeys has been discovered, a person does not need to find textual evidence to know that this new monkey, too, will have legs (a meronymic relationship inferred due to a hyponymic relation to monkeys in general).

We introduce a model that can accurately predict additional true facts using only an existing database. This is achieved by representing each entity (i.e., each object or individual) in the database as a vector. These vectors can capture facts about that entity and how probable it is part of a certain relation. Each relation is defined through the parameters of a novel neural tensor network which can explicitly relate two entity vectors. The first contribution of this paper is the new neural tensor network (NTN), which generalizes several previous neural network models and provides a more powerful way to model relational information than a standard neural network layer.

The second contribution is to introduce a new way to represent entities in knowledge bases. Previous work [8, 9, 10] represents each entity with one vector. However, does not allow the sharing of

---

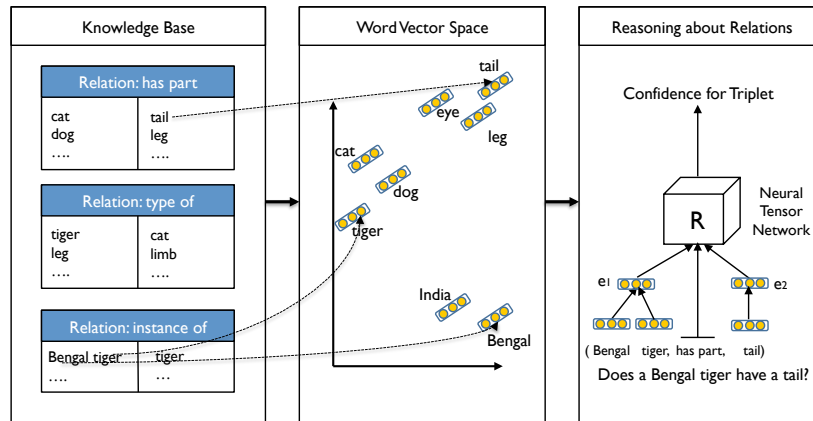

Figure 1: Overview of our model which learns vector representations for entries in a knowledge base in order to predict new relationship triples. If combined with word representations, the relationships can be predicted with higher accuracy and for entities that were not in the original knowledge base.

statistical strength if entity names share similar substrings. Instead, we represent each entity as the average of its word vectors, allowing the sharing of statistical strength between the words describing each entity e.g., *Bank of China* and *China*.

The third contribution is the incorporation of word vectors which are trained on large unlabeled text. This readily available resource enables all models to more accurately predict relationships.

We train on relationships in WordNet and Freebase and evaluate on a heldout set of unseen relational triplets. Our model outperforms previously introduced related models such as those of [8, 9, 10]. Our new model, illustrated in Fig. 1, outperforms previous knowledge base models by a large margin. We will make the code and dataset available at `www.socher.org`.

## 2   Related Work

The work most similar to ours is that by Bordes et al. [8] and Jenatton et al. [9] who also learn vector representations for entries in a knowledge base. We implement their approach and compare to it directly. Our new model outperforms this and other previous work. We also show that both our and their model can benefit from initialization with unsupervised word vectors.

Another related approach is by Sutskever et al. [11] who use tensor factorization and Bayesian clustering for learning relational structures. Instead of clustering the entities in a nonparametric Bayesian framework we rely purely on learned entity vectors. Their computation of the truth of a relation can be seen as a special case of our proposed model. Instead of using MCMC for inference and learning, we use standard forward propagation and backpropagation techniques modified for the NTN. Lastly, we do not require multiple embeddings for each entity. Instead, we consider the subunits (space separated words) of entity names.

Our Neural Tensor Network is related to other models in the deep learning literature. Ranzato and Hinton [12] introduced a factored 3-way Restricted Boltzmann Machine which is also parameterized by a tensor. Recently, Yu et al. [13] introduce a model with tensor layers for speech recognition. Their model is a special case of our model and is only applicable inside deeper neural networks. Simultaneously with this paper, we developed a recursive version of this model for sentiment analysis [14].

There is a vast amount of work on extending knowledge bases by parsing external, text corpora [5, 6, 2], among many others. The field of open information extraction [15], for instance, extracts relationships from millions of web pages. This work is complementary to ours; we mainly note that little work has been done on knowledge base extension based purely on the knowledge base itself or with readily available resources but without re-parsing a large corpus.

Lastly, our model can be seen as learning a tensor factorization, similar to Nickel et al. [16]. In the comparison of Bordes et al. [17] these factorization methods have been outperformed by energy-based models.

Many methods that use knowledge bases as features such as [3, 4] could benefit from a method that maps the provided information into vector representations. We learn to modify word representations via grounding in world knowledge. This essentially allows us to analyze word embeddings and query them for specific relations. Furthermore, the resulting vectors could be used in other tasks such as named entity recognition [18] or relation classification in natural language [19].

# 3 Neural Models for Reasoning over Relations

This section introduces the neural tensor network that reasons over database entries by learning vector representations for them. As shown in Fig. 1, each relation triple is described by a neural network and pairs of database entities which are given as input to that relation's model. The model returns a high score if they are in that relationship and a low one otherwise. This allows any fact, whether implicit or explicitly mentioned in the database to be answered with a certainty score. We first describe our neural tensor model and then show that many previous models are special cases of it.

## 3.1 Neural Tensor Networks for Relation Classification

The goal is to learn models for common sense reasoning, the ability to realize that some facts hold purely due to other existing relations. Another way to describe the goal is link prediction in an existing network of relationships between entity nodes. The goal of our approach is to be able to state whether two entities $(e_1, e_2)$ are in a certain relationship $R$. For instance, whether the relationship $(e_1, R, e_2) = ($*Bengal tiger*, *has part*, *tail*$)$ is true and with what certainty. To this end, we define a set of parameters indexed by $R$ for each relation's scoring function. Let $e_1, e_2 \in \mathbb{R}^d$ be the vector representations (or features) of the two entities. For now we can assume that each value of this vector is randomly initialized to a small uniformly random number.

The Neural Tensor Network (NTN) replaces a standard linear neural network layer with a bilinear tensor layer that directly relates the two entity vectors across multiple dimensions. The model computes a score of how likely it is that two entities are in a certain relationship by the following NTN-based function:

$$g(e_1, R, e_2) = u_R^T f\left(e_1^T W_R^{[1:k]} e_2 + V_R \begin{bmatrix} e_1 \\ e_2 \end{bmatrix} + b_R\right), \qquad (1)$$

where $f = \tanh$ is a standard nonlinearity applied element-wise, $W_R^{[1:k]} \in \mathbb{R}^{d \times d \times k}$ is a tensor and the bilinear tensor product $e_1^T W_R^{[1:k]} e_2$ results in a vector $h \in \mathbb{R}^k$, where each entry is computed by one slice $i = 1, \ldots, k$ of the tensor: $h_i = e_1^T W_R^{[i]} e_2$. The other parameters for relation $R$ are the standard form of a neural network: $V_R \in \mathbb{R}^{k \times 2d}$ and $U \in \mathbb{R}^k, b_R \in \mathbb{R}^k$.

Fig. 2 shows a visualization of this model. The main advantage is that it can relate the two inputs multiplicatively instead of only implicitly through the nonlinearity as with standard neural networks where the entity vectors are simply concatenated. Intuitively, we can see each slice of the tensor as being responsible for one type of entity pair or instantiation of a relation. For instance, the model could learn that both animals and mechanical entities such as cars can have parts (i.e., (car, has part, $x$)) from different parts of the semantic word vector space. In our experiments, we show that this results in improved performance. Another way to interpret each tensor slice is that it mediates the relationship between the two entity vectors differently.

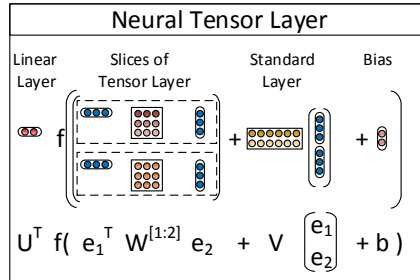

Figure 2: Visualization of the Neural Tensor Network. Each dashed box represents one slice of the tensor, in this case there are $k = 2$ slices.

## 3.2 Related Models and Special Cases

We now introduce several related models in increasing order of expressiveness and complexity. Each model assigns a score to a triplet using a function $g$ measuring how likely the triplet is correct. The ideas and strengths of these models are combined in our new Neural Tensor Network defined above.

**Distance Model.** The model of Bordes et al. [8] scores relationships by mapping the left and right entities to a common space using a relationship specific mapping matrix and measuring the $L_1$ distance between the two. The scoring function for each triplet has the following form:

$$g(e_1, R, e_2) = \|W_{R,1}e_1 - W_{R,2}e_2\|_1,$$

where $W_{R,1}, W_{R,2} \in \mathbb{R}^{d \times d}$ are the parameters of relation $R$'s classifier. This similarity-based model scores correct triplets lower (entities most certainly in a relation have 0 distance). All other functions are trained to score correct triplets higher. The main problem with this model is that the parameters of the two entity vectors do not interact with each other, they are independently mapped to a common space.

**Single Layer Model.** The second model tries to alleviate the problems of the distance model by connecting the entity vectors implicitly through the nonlinearity of a standard, single layer neural network. The scoring function has the following form:

$$g(e_1, R, e_2) = u_R^T f\left(W_{R,1}e_1 + W_{R,2}e_2\right) = u_R^T f\left([W_{R,1} W_{R,2}] \begin{bmatrix} e_1 \\ e_2 \end{bmatrix}\right),$$

where $f = \tanh$, $W_{R,1}, W_{R,2} \in \mathbb{R}^{k \times d}$ and $u_R \in \mathbb{R}^{k \times 1}$ are the parameters of relation $R$'s scoring function. While this is an improvement over the distance model, the non-linearity only provides a weak interaction between the two entity vectors at the expense of a harder optimization problem. Collobert and Weston [20] trained a similar model to learn word vector representations using words in their context. This model is a special case of the tensor neural network if the tensor is set to 0.

**Hadamard Model.** This model was introduced by Bordes et al. [10] and tackles the issue of weak entity vector interaction through multiple matrix products followed by Hadamard products. It is different to the other models in our comparison in that it represents each relation simply as a single vector that interacts with the entity vectors through several linear products all of which are parameterized by the same parameters. The scoring function is as follows:

$$g(e_1, R, e_2) = (W_1 e_1 \otimes W_{rel,1} e_R + b_1)^T (W_2 e_2 \otimes W_{rel,2} e_R + b_2)$$

where $W_1, W_{rel,1}, W_2, W_{rel,2} \in \mathbb{R}^{d \times d}$ and $b_1, b_2 \in \mathbb{R}^{d \times 1}$ are parameters that are shared by *all* relations. The only relation specific parameter is $e_R$. While this allows the model to treat relational words and entity words the same way, we show in our experiments that giving each relationship its own matrix operators results in improved performance. However, the bilinear form between entity vectors is by itself desirable.

**Bilinear Model.** The fourth model [11, 9] fixes the issue of weak entity vector interaction through a relation-specific bilinear form. The scoring function is as follows: $g(e_1, R, e_2) = e_1^T W_R e_2$, where $W_R \in \mathbb{R}^{d \times d}$ are the only parameters of relation $R$'s scoring function. This is a big improvement over the two previous models as it incorporates the interaction of two entity vectors in a simple and efficient way. However, the model is now restricted in terms of expressive power and number of parameters by the word vectors. The bilinear form can only model linear interactions and is not able to fit more complex scoring functions. This model is a special case of NTNs with $V_R = 0, b_R = 0, k = 1, f = $ identity. In comparison to bilinear models, the neural tensor has much more expressive power which will be useful especially for larger databases. For smaller datasets the number of slices could be reduced or even vary between relations.

## 3.3 Training Objective and Derivatives

All models are trained with contrastive max-margin objective functions. The main idea is that each triplet in the training set $T^{(i)} = (e_1^{(i)}, R^{(i)}, e_2^{(i)})$ should receive a higher score than a triplet in which one of the entities is replaced with a random entity. There are $N_R$ many relations, indexed by $R^{(i)}$ for each triplet. Each relation has its associated neural tensor net parameters. We call the triplet

with a random entity corrupted and denote the corrupted triplet as $T_c^{(i)} = (e_1^{(i)}, R^{(i)}, e_c)$, where we sampled entity $e_c$ randomly from the set of all entities that can appear at that position in that relation. Let the set of all relationships' NTN parameters be $\mathbf{\Omega} = \mathbf{u}, \mathbf{W}, \mathbf{V}, \mathbf{b}, \mathbf{E}$. We minimize the following objective:

$$J(\mathbf{\Omega}) = \sum_{i=1}^{N} \sum_{c=1}^{C} \max\left(0, 1 - g\left(T^{(i)}\right) + g\left(T_c^{(i)}\right)\right) + \lambda \|\mathbf{\Omega}\|_2^2,$$

where $N$ is the number of training triplets and we score the correct relation triplet higher than its corrupted one up to a margin of 1. For each correct triplet we sample $C$ random corrupted triplets. We use standard $L_2$ regularization of all the parameters, weighted by the hyperparameter $\lambda$.

The model is trained by taking derivatives with respect to the five groups of parameters. The derivatives for the standard neural network weights $V$ are the same as in general backpropagation. Dropping the relation specific index $R$, we have the following derivative for the $j$'th slice of the full tensor:

$$\frac{\partial g(e_1, R, e_2)}{\partial W^{[j]}} = u_j f'(z_j) e_1 e_2^T, \quad \text{where} \quad z_j = e_1^T W^{[j]} e_2 + V_j \cdot \begin{bmatrix} e_1 \\ e_2 \end{bmatrix} + b_j,$$

where $V_j$ is the $j$'th row of the matrix $V$ and we defined $z_j$ as the $j$'th element of the $k$-dimensional hidden tensor layer. We use minibatched L-BFGS for optimization which converges to a local optimum of our non-convex objective function. We also experimented with AdaGrad but found that it performed slightly worse.

## 3.4 Entity Representations Revisited

All the above models work well with randomly initialized entity vectors. In this section we introduce two further improvements: representing entities by their word vectors and initializing word vectors with pre-trained vectors.

Previous work [8, 9, 10] assigned a single vector representation to each entity of the knowledge base, which does not allow the sharing of statistical strength between the words describing each entity. Instead, we model each word as a $d$-dimensional vector $\in \mathbb{R}^d$ and compute an entity vector as the *composition* of its word vectors. For instance, if the training data includes a fact that *homo sapiens* is a type of *hominid* and this entity is represented by two vectors $v_{homo}$ and $v_{sapiens}$, we may extend the fact to the previously unseen *homo erectus*, even though its second word vector for *erectus* might still be close to its random initialization.

Hence, for a total number of $N_E$ entities consisting of $N_W$ many unique words, if we train on the word level (the training error derivatives are also back-propagated to these word vectors), and represent entities by word vectors, the full embedding matrix has dimensionality $E \in \mathbb{R}^{d \times N_W}$. Otherwise we represent each entity as an atomic single vector and train the entity embedding matrix $E \in \mathbb{R}^{d \times N_E}$.

We represent the entity vector by averaging its word vectors. For example, $v_{homo\ sapiens} = 0.5(v_{homo} + v_{sapiens})$. We have also experimented with Recursive Neural Networks (RNNs) [21, 19] for the composition. In the WordNet subset over 60% of the entities have only a single word and over 90% have less or equal to 2 words. Furthermore, most of the entities do not exhibit a clear compositional structure, e.g., people names in Freebase. Hence, RNNs did not show any distinct improvement over simple averaging and we will not include them in the experimental results.

Training word vectors has the additional advantage that we can benefit from pre-trained unsupervised word vectors, which in general capture some distributional syntactic and semantic information. We will analyze how much it helps to use these vectors for initialization in Sec. 4.2. Unless otherwise specified, we use the $d = 100$-dimensional vectors provided by [18]. Note that our approach does not explicitly deal with polysemous words. One possible future extension is to incorporate the idea of multiple word vectors per word as in Huang et al. [22].

## 4 Experiments

Experiments are conducted on both WordNet [1] and FreeBase [23] to predict whether some relations hold using other facts in the database. This can be seen as common sense reasoning [24] over known facts or link prediction in relationship networks. For instance, if somebody was born

in *London*, then their nationality would be *British*. If a *German Shepard* is a *dog*, it is also a *vertebrate*. Our models can obtain such knowledge (with varying degrees of accuracy) by jointly learning relationship classifiers and entity representations.

We first describe the datasets, then compare the above models and conclude with several analyses of important modeling decisions, such as whether to use entity vectors or word vectors.

## 4.1 Datasets

| Dataset | #R. | # Ent. | # Train | # Dev | # Test |
|---|---|---|---|---|---|
| Wordnet | 11 | 38,696 | 112,581 | 2,609 | 10,544 |
| Freebase | 13 | 75,043 | 316,232 | 5,908 | 23,733 |

Table 1: The statistics for WordNet and Freebase including number of different relations #R.

Table 1 gives the statistics of the databases. For WordNet we use 112,581 relational triplets for training. In total, there are 38,696 unique entities in 11 different relations. One important difference to previous work is our dataset generation which filters trivial test triplets. We filter out tuples from the testing set if either or both of their two entities also appear in the training set in a different relation or order. For instance, if $(e_1, similar\ to, e_2)$ appears in training set, we delete $(e_2, similar\ to, e_1)$ and $(e_1, type\ of, e_2)$, etc from the testing set. In the case of synsets containing multiple words, we pick the first, most frequent one. For FreeBase, we use the relational triplets from *People* domain, and extract 13 relations. We remove 6 of them (*place of death, place of birth, location, parents, children, spouse*) from the testing set since they are very difficult to predict, e.g., the name of somebody's spouse is hard to infer from other knowledge in the database.

It is worth noting that the setting of FreeBase is profoundly different from WordNet's. In WordNet, $e_1$ and $e_2$ can be arbitrary entities; but in FreeBase, $e_1$ is restricted to be a person's name, and $e_2$ can only be chosen from a finite answer set. For example, if $R = gender$, $e_2$ can only be *male* or *female*; if $R = nationality$, $e_2$ can only be one of $188$ country names. All the relations for testing and their answer set sizes are shown in Fig. 3.

We use a different evaluation set from [8] because it has become apparent to us and them that there were issues of overlap between their training and testing sets which impacted the quality and interpretability of their evaluation.

## 4.2 Relation Triplets Classification

Our goal is to predict correct facts in the form of relations $(e_1, R, e_2)$ in the testing data. This could be seen as answering questions such as *Does a dog have a tail?*, using the scores g(dog, has part, tail) computed by the various models.

We use the development set to find a threshold $T_R$ for each relation such that if $g(e_1, R, e_2) \geq T_R$, the relation $(e_1, R, e_2)$ holds, otherwise it does not hold.

In order to create a testing set for classification, we randomly switch entities from correct testing triplets resulting in a total of $2 \times \#\text{Test}$ triplets with equal number of positive and negative examples. In particular, we constrain the entities from the possible answer set for Freebase by only allowing entities in a position if they appeared in that position in the dataset. For example, given a correct triplet (*Pablo Picaso*, *nationality*, *Spain*), a potential negative example is (*Pablo Picaso*, *nationality*, *United States*). We use the same way to generate the development set. This forces the model to focus on harder cases and makes the evaluation harder since it does not include obvious non-relations such as (*Pablo Picaso*, *nationality*, *Van Gogh*). The final accuracy is based on how many triplets are classified correctly.

**Model Comparisons**
We first compare the accuracy among different models. In order to get the highest accuracy for all the models, we cross-validate using the development set to find the best hyperparameters: (i) vector initialization (see next section); (ii) regularization parameter $\lambda = 0.0001$; (iii) the dimensionality of the hidden vector (for the single layer and NTN models $d = 100$) and (iv) number of training iterations $T = 500$. Finally, the number of slices was set to 4 in our NTN model.

Table 2 shows the resulting accuracy of each model. Our Neural Tensor Network achieves an accuracy of 86.2% on the Wordnet dataset and 90.0% on Freebase, which is at least $2\%$ higher than the bilinear model and $4\%$ higher than the Single Layer Model.

| Model | WordNet | Freebase | Avg. |
|---|---|---|---|
| Distance Model | 68.3 | 61.0 | 64.7 |
| Hadamard Model | 80.0 | 68.8 | 74.4 |
| Single Layer Model | 76.0 | 85.3 | 80.7 |
| Bilinear Model | 84.1 | 87.7 | 85.9 |
| Neural Tensor Network | **86.2** | **90.0** | **88.1** |

Table 2: Comparison of accuracy of the different models described in Sec. 3.2 on both datasets.

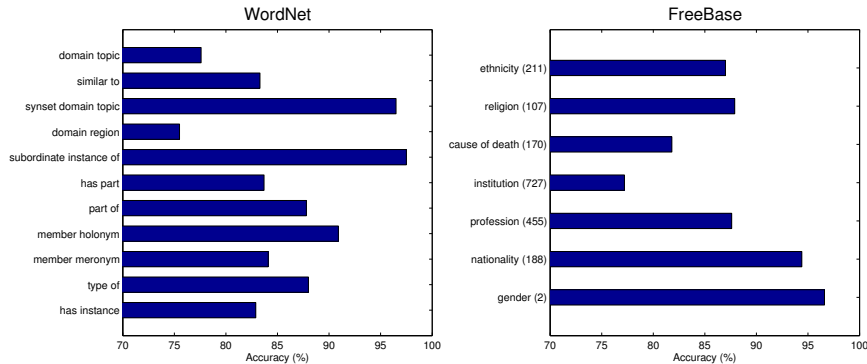

Figure 3: Comparison of accuracy of different relations on both datasets. For FreeBase, the number in the bracket denotes the size of possible answer set.

First, we compare the accuracy among different relation types. Fig. 3 reports the accuracy of each relation on both datasets. Here we use our NTN model for evaluation, other models generally have a lower accuracy and a similar distribution among different relations. The accuracy reflects the difficulty of inferring a relationship from the knowledge base.

On WordNet, the accuracy varies from 75.5% (*domain region*) to 97.5% (*subordinate instance of*). Reasoning about some relations is more difficult than others, for instance, the relation (*dramatic art*, *domain region*, *closed circuit television*) is much more vague than the relation (*missouri, subordinate instance of, river*). Similarly, the accuracy varies from 77.2% (*institution*) to 96.6% (*gender*) in FreeBase. We can see that the two easiest relations for reasoning are *gender* and *nationality*, and the two most difficult ones are *institution* and *cause of death*. Intuitively, we can infer the *gender* and *nationality* from the *name*, *location*, or *profession* of a person, but we hardly infer a person's cause of death from all other information.

We now analyze the choice of entity representations and also the influence of word initializations. As explained in Sec. 3.4, we compare training entity vectors ($E \in \mathbb{R}^{d \times N_E}$) and training word vectors ($E \in \mathbb{R}^{d \times N_W}$), where an entity vector is computed as the average of word vectors. Furthermore, we compare random initialization and unsupervised initialization for training word vectors. In summary, we explore three options: (i) entity vectors (EV); (ii) randomly initialized word vectors (WV); (iii) word vectors initialized with unsupervised word vectors (WV-init).

Fig. 4 shows the various models and their performance with these three settings. We observe that word vectors consistently and significantly outperform entity vectors on WordNet and this also holds in most cases on FreeBase. It might be because the entities in WordNet share more common words. Furthermore, we can see that most of the models have improved accuracy with initialization from unsupervised word vectors. Even with random initialization, our NTN model with training word vectors can reach high classification accuracy: 84.7% and 88.9% on WordNet and Freebase respectively. In other words, our model is still able to perform good reasoning without external textual resources.

## 4.3 Examples of Reasoning

We have shown that our model can achieve high accuracy when predicting whether a relational triplet is true or not. In this section, we give some example predictions. In particular, we are interested in how the model does transitive reasoning across multiple relationships in the knowledge base.

First, we demonstrate examples of relationship predictions by our Neural Tensor Network on Word-Net. We select the first entity and a relation and then sort all the entities (represented by their word

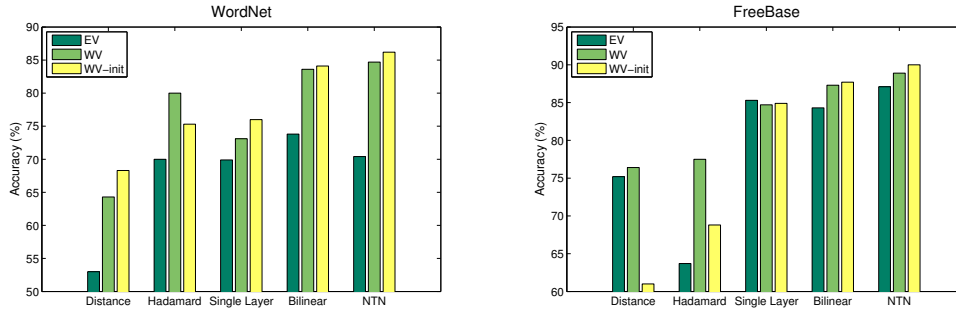

Figure 4: Influence of entity representations. **EV**: entity vectors. **WV**: randomly initialized word vectors. **WV-init**: word vectors initialized with unsupervised semantic word vectors.

| Entity $e_1$ | Relationship $R$ | Sorted list of entities likely to be in this relationship |
|---|---|---|
| tube | type of | structure; anatomical structure; device; body; body part; organ |
| creator | type of | individual; adult; worker; man; communicator; instrumentalist |
| dubrovnik | subordinate instance of | city; town; city district; port; river; region; island |
| armed forces | domain region | military operation; naval forces; military officer; military court |
| boldness | has instance | audaciousness; aggro; abductor; interloper; confession; |
| peole | type of | group; agency; social group; organisation; alphabet; race |

Table 3: Examples of a ranking by the model for right hand side entities in WordNet. The ranking is based on the scores that the neural tensor network assigns to each triplet.

vector averages) by descending scores that the model assigns to the complete triplet. Table 3 shows some examples for several relations, and most of the inferred relations among them are plausible.

Fig. 5 illustrates a real example from FreeBase in which a person's information is inferred from the other relations provided in training. Given *place of birth* is *Florence* and *profession* is *historian*, our model can accurately predict that *Francesco Guicciardini*'s *gender* is *male* and his *nationality* is *Italy*. These might be infered from two pieces of common knowledge: (i) *Florence* is a city of *Italy*; (ii) *Francesco* is a common name among males in *Italy*. The key is how our model can derive these facts from the knowledge base itself, without the help of external information. For the first fact, some relations such as *Matteo Rosselli* has *location Florence* and *nationality Italy* exist in the knowledge base, which might imply the connection between *Florence* and *Italy*. For the second fact, we can see that many other people e.g., *Francesco Patrizi* are shown *Italian* or *male* in the FreeBase, which might imply that *Francesco* is a male or Italian name. It is worth not-

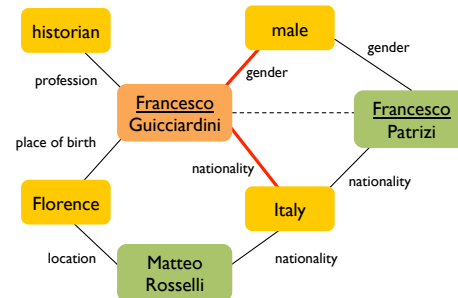

Figure 5: A reasoning example in FreeBase. Black lines denote relationships given in training, red lines denote relationships the model inferred. The dashed line denotes word vector sharing.

ing that we do not have an explicit relation between *Francesco Guicciardini* and *Francesco Patrizi*; the dashed line in Fig. 5 shows the benefits from the sharing via word representations.

## 5   Conclusion

We introduced Neural Tensor Networks for knowledge base completion. Unlike previous models for predicting relationships using entities in knowledge bases, our model allows mediated interaction of entity vectors via a tensor. The model obtains the highest accuracy in terms of predicting unseen relationships between entities through reasoning inside a given knowledge base. It enables the extension of databases even without external textual resources. We further show that by representing entities through their constituent words and initializing these word representations using readily available word vectors, performance of all models improves substantially. Potential path for future work include scaling the number of slices based on available training data for each relation and extending these ideas to reasoning over free text.

**Acknowledgments**

Richard is partly supported by a Microsoft Research PhD fellowship. The authors gratefully acknowledge the support of a Natural Language Understanding-focused gift from Google Inc., the Defense Advanced Research Projects Agency (DARPA) Deep Exploration and Filtering of Text (DEFT) Program under Air Force Research Laboratory (AFRL) prime contract no. FA8750-13-2-0040, the DARPA Deep Learning program under contract number FA8650-10-C-7020 and NSF IIS-1159679. Any opinions, findings, and conclusions or recommendations expressed in this material are those of the authors and do not necessarily reflect the view of DARPA, AFRL, or the US government.

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
