[Reviews · NeurIPS 2013]

Submitted by Assigned_Reviewer_4

This paper proposes a new model for link prediction in knowledge bases. By learning a non-linear scoring function based on tensor and matrix operators for representing relations and word embeddings for representing entities, this approach can outperform previous work on data from two knowledge bases: Wordnet and Freebase.



Clarity: The paper is very clear and nicely written. I would only reduce the use of the term "reasoning", which is not obviously justified in this context and can be misleading. It seems that link prediction is more performed using (well-trained) similarity measures than by relying on (formal) reasoning.

Originality: While building on previous work (cited), the model architecture is original and the use of word embeddings (learned on text corpora) is also brand new.

Significance: The proposed approach allows to outperform previous work on two datasets. According to experimental results, the effect of using word embeddings seems to be much more important than the neural tensor architecture. However, it is disappointing that only embedding-related methods are compared (and discussed). Comparisons with previous work should be improved.

For instance, tensor factorization has been the first kind of methods applied to this kind of problem. And recent work by Nickel et al. (e.g. RESCAL at ICML11 and WWW12) has shown that such approaches were efficient for link prediction, even when the number of entities is large. Comparison is not provided and the data sets and evaluation protocols proposed here differ from those that have been used in previous work. There was a problem with data from [7] but at least [8,10] have been evaluated on existing and public data sets (so did RESCAL).

Quality: The paper is of good quality, but suffers from these problems regarding references and evaluation.

It seems that the use of word embeddings is the key part to improve performance. But, such embeddings might not be available for many knowledges bases, concerning specialized topic such as biology or user modeling, for which no corresponding text data is available. This approach is then mostly useful for a reduced set of knowledge bases (which can be still quite large). However, claiming that the goal of the model is to "accurately predict additional true facts using only an existing database" seems to be slightly in contradiction with the fact of relying on associated text data to learn the word embeddings (even if this text is unlabeled).

** after rebuttal **
I am satisfied by the rebuttal except for the answer to the above comment. On WordNet w/o WV, NTN is outperformed by the Bilinear and (almost) tied by Hadamard and Single Layer (Fig. 4 left).
** **

Concerning Wordnet, it is not clear how the words are mapped to the knowledge base nodes. Wordnet nodes are synset (i.e. sets of words sharing the same sense). Many synsets contain a single word but the most frequent ones contain several synonyms. Which word was chosen? How was polysemy dealt with?

Summary: This paper proposes a new model for link prediction in knowledge bases, which can fruitfully use database information and unlabeled text data. The paper is clear and the method original. However, some comparisons and key references to previous work are lacking.

Submitted by Assigned_Reviewer_6

This paper describes a model for multi-relational data specialized to knowledge bases and other datasets with entities and relationships labeled by words. The model is a straightforward generalization of other models from the literature. The model uses entity vectors that are the average of word vectors in the entity name. The score of a triple (e1, R, e2) is based on three vector-valued terms fed through an elementwise nonlinearity and then dotted with a relation-dependent vector. One term is a bias vector that is a function of R. One term has a weight matrix dependent on R applied to the concatenation of the vectors for e1 and e2. The final term is a bilinear tensor product using a relation-dependent set of weights and the two entity vectors. To justify the use of this model, the paper presents results on classifying (entity, relation, entity) triples as positive (from a knowledge base) or negative (synthetic triples unlikely to exist in the knowledge base).

I don't see the advantage of the non-linearity in equation 1. I expect the model would be mostly unchanged in practice if f was removed from equation 1 (and u_R was folded into other parameters where appropriate). When all the embedding vectors are learned and unconstrained and there are so many arbitrary relation-dependent weights the nonlinearity adds very little power to the model since the mapping from entity id to entity vector is arbitrary.

Although the model in the paper is powerful and potentially interesting, the experimental evaluation has a few serious weaknesses. The full train/dev/test splitting protocal needs to be described precisely. If (x,r,y) is in the test set, can (x,r, z != y) be in the training set? Or can only (u!=x, r, v!=y) and (x, q != r, y) be in the training set? The most powerful models should do best when they have lots of data and have seen entities in the test set many times and seen relations in the test set many times. Therefore the most interesting experiments for the model described in the paper are ones that test the generalization ability of the model from limited training data. Along these lines, why are results on all relations in Freebase relegated to the supplementary material and so few relations used in the main paper? A dozen relations is simply not enough to do any meaningful evaluation of this model, especially since it has so many relation-specific parameters. And how many entities are there for the full experiments? If anything, remove the experiments that focus on the small subset of Freebase and if necessary break down some of the results based on how much relevant training data was used for the particular test cases. The model needs to be tested in the regime where it has very little data for the particular relation being tested and in the regime where it has very little data for the entities; that is where simpler special cases of the model may shine. My own view is that a simpler model will probably work better in most situations although it is possible we as a community have not yet settled on the best simple model. There are many obvious extensions to models in the literature that add modeling power and when a paper introduces one it must demonstrate that the overfitting risks can be controlled and that the new model is actually really useful.

Figure 3 demonstrates what is probably an overfitting effect for the NTN model on WordNet since the NTN does so poorly when trained without word vectors. This supports my concern that overfitting is a larger than usual risk with this model and that the best evaluation would employ data with relations that have few training cases associated with them.

The explanation of how negative cases were generated for the classification task needs more detail. Can the negative triple (Pablo Picasso, nationality, George Washington) get generated?

I don't understand why a classification task was used instead of a ranking or information retrieval style task as in [7]. Using a classification task seems non-standard and makes it harder to compare to other papers. Also for open domain knowledge base completion, a ranking task makes more sense.

I recommend removing the average column of table 2 since it adds nothing and averaging results across tasks should be avoided.
Summary: The paper describes a new and powerful model for a very important task, but does not provide a maximally compelling empirical evaluation.

Submitted by Assigned_Reviewer_7

This paper proposes a model of entity relationships to predict relationships for new entity pairs. It is an extension of a neural network model that allows multiple linear interactions between each pair of features between entity vectors. Entity vectors are further improved using compositions of word vectors. Empirical evaluation shows significant improvements for both these novel contributions.

This is an elegant and powerful model of entity relationships, and it performs well. The model of the composition of words in an entity name is simple, but enough to indicate the importance of this issue for this problem. Some improvement is also shown from using textual resources to initialize the word vectors, again showing the relevance of another line of research to this problem. The paper includes comparisons to 4 other simpler models, justifying the power of this model.

The writing and presentation are clear, and the paper includes examples of relations and to illustrate the kind of inference the model is able to do.
Summary: This is an elegant and powerful model of entity relationships, and it performs well. The importance of compositional representations of entity vectors is also shown.
Author Feedback

Author rebuttal: We thank the reviewers for the very insightful reviews.


@Reviewer_4
Thank you for the pointer to Nickel et al.’s “A Three-Way Model for Collective Learning on Multi-Relational Data”, this is indeed relevant and we will add a comparison to their model.
In the related work section of paper “Irreflexive and Hierarchical Relations as Translations”, Bordes et al. mentioned that “models based on tensor factorization (e.g. Nickel et al. 2011) have shown to be efficient. However, they have been outperformed by energy-based models.”

In the paper “A latent factor model for highly multi-relational data”, they compared their model and RESCAL, and showed that their approach outperforms RESCAL on 2 of 3 datasets. We outperform their model so it is likely the new experiments will show that we can also outperform RESCAL.

“the effect of using word embeddings seems to be much more important than the neural tensor architecture.”

This conclusion is not supported by the data in the paper. Fig. 4 (right) shows that the improvement comes in roughly equal parts.We do find that the embeddings learned on text corpora improve all methods.

“There was a problem with data from [7] but at least[8,10] have been evaluated on existing and public data sets (so didRESCAL).”

The datasets used in [8,10] are purpose-specific and extremely small –the kinship dataset contains 26 kinship relations among 104 people. Our used WordNet and FreeBase datasets have 38,696 entities and 75,043 entities respectively, and the entities / relations are widespread. Thus we aim to predict true facts for general and large-scale databases, and this result is more significant.

“How would the complete model perform?”
The addition of the single layer to [8] did not improve the model.

Which word was chosen [from a WordNetsynset]?
The most frequent, first one.

How was polysemy dealt with?
The embeddings seem to be able to capture this; so polysemous words as subunits of entities use the same vector for sharing additional statistical strength.

“But, such embeddings might not be available for many knowledges bases, concerning specialized topic such as biology or user modeling, for which no corresponding text data is available. This approach is then mostly useful for a reduced set of knowledge bases(which can be still quite large). However, claiming that the goal of the model is to "accurately predict additional true facts using onlyan existing database" seems to be slightly in contradiction with thefact of relying on associated text data to learn the word embeddings(even if this text is unlabeled).”

Even if we don’t use the unlabeled text data for initializing word embeddings (using random initialization instead), we can still reach very high accuracies (84.7% for WordNet and 88.9% for FreeBase), thus our model is able to perform good reasoning without external textual resources.


@Reviewer_6
“I expect the model would be mostly unchanged in practice if f was removed from equation 1”
The nonlinearity does help, we will add the comparison to the supplementary material.
“If (x,r,y) is in the test set, can (x,r, z != y) be in thetraining set? Or can only (u!=x, r, v!=y) and (x, q != r, y) be in thetraining set?”

That is correct, we constructed this dataset to be much harder by not even allowing “(x,r, z != y)”, since there are symmetric relationships such as (x , is similar to, y).
However, (x, r, z != y) is possible in the training set for WordNet if (x, r, ?) has multiple answers, but it is impossible for FreeBase.

(x, q != r, y) cannot be in the training set, since we removed all triples (x, q, y) or (y, q, x) (q can be any relation) from the training set if (x, r, y) is in the testing set, which made this dataset much harder.

“Therefore the most interesting experiments for the model described in the paper are ones that test the generalization ability of the model from limited training data.”

We do see a need for powerful models that can deal well with large training datasets in this age of “big data”

We filtered the relationships using almost the same setup as previous work by Bordes et al.

“it must demonstrate that [...] the new model is actually really useful.”
We demonstrate this in the experiments. We also propose an improvement to ALL previous embedding based models via the word vector learning.

Fig. 3 shows that the NTN outperforms all other models even with random word vectors, it does not show an overfitting problem.

We do share your concern of the powerful model possibly overfitting when little data is available. There are easy fixes for the NTN available to mitigate this. For instance, the number of tensor slices for each relationship could be made dependent on the number of training samples for that relationship. We can explore this to further improve the model performance.


Can the negative triple (Pablo Picasso, nationality, George Washington) get generated?
Section 4.2 answers this. “In particular, we constrain the entities from the possible answer set for Freebase”, we can only generate negative triples (Pablo Picasso, nationality, SOME COUNTRY).